# LAR receptor phospho-tyrosine phosphatases regulate NMDA-receptor responses

**Alessandra Sclip\*, Thomas C Südhof**

Department of Cellular and Molecular Physiology, Howard Hughes Medical Institute, Stanford University School of Medicine, Stanford, United States

**Abstract** LAR-type receptor phosphotyrosine-phosphatases (LAR-RPTPs) are presynaptic adhesion molecules that interact trans-synaptically with multitudinous postsynaptic adhesion molecules, including SliTrks, SALMs, and TrkC. Via these interactions, LAR-RPTPs are thought to function as synaptogenic wiring molecules that promote neural circuit formation by mediating the establishment of synapses. To test the synaptogenic functions of LAR-RPTPs, we conditionally deleted the genes encoding all three LAR-RPTPs, singly or in combination, in mice before synapse formation. Strikingly, deletion of LAR-RPTPs had no effect on synaptic connectivity in cultured neurons or in vivo, but impaired NMDA-receptor-mediated responses. Deletion of LAR-RPTPs decreased NMDA-receptor-mediated responses by a trans-synaptic mechanism. In cultured neurons, deletion of all LAR-RPTPs led to a reduction in synaptic NMDA-receptor EPSCs, without changing the subunit composition or the protein levels of NMDA-receptors. In vivo, deletion of all LAR-RPTPs in the hippocampus at birth also did not alter synaptic connectivity as measured via AMPA-receptor-mediated synaptic responses at Schaffer-collateral synapses monitored in juvenile mice, but again decreased NMDA-receptor mediated synaptic transmission. Thus, LAR-RPTPs are not essential for synapse formation, but control synapse properties by regulating postsynaptic NMDA-receptors via a trans-synaptic mechanism that likely involves binding to one or multiple postsynaptic ligands.

**\*For correspondence:**
asclip@stanford.edu

**Competing interests:** The authors declare that no competing interests exist.

## Introduction

In the brain, neurons wire to form distinct neural circuits that are important for processing information. Neural circuit wiring requires growth of axons towards target regions, self-avoidance of axons and dendrites, axon-dendrite target selection, and formation and specification of synapses (*Hassan and Hiesinger, 2015*; *Yogev and Shen, 2014*; *Zipursky and Sanes, 2010*). The molecular mechanisms governing synapse formation and specification remain largely unknown.

Synaptic cell-adhesion molecules form trans-synaptic complexes that are thought to initiate synapse formation, maintain synapse stability, and regulate synapse properties (*Südhof, 2018*). Multiple synaptic cell-adhesion molecules have been identified, among which LAR-RPTPs (a.k.a. type IIA PTPRs) are prominent because strong evidence suggests that they are key drivers of synapse formation (*Joo et al., 2014*; *Kwon et al., 2010*; *Li et al., 2015*; *Takahashi et al., 2011*; *Um et al., 2014*; *Yim et al., 2013*). LAR-RPTPs are type I transmembrane proteins encoded by three genes in vertebrates (PTPRS, PTPRD, and PTPRF) (*Pulido et al., 1995*). LAR-RPTPs are composed of a large extracellular sequence containing three immunoglobulin-like (Ig) domains and multiple (4-8) fibronectin-3 (FN3) repeats, and of a similarly large intracellular sequence containing two phospho-tyrosine phosphatase (PTP) domains (*Figure 1A*). LAR-RPTPs mediate essential functions in and outside of the brain during development (*Elchebly et al., 1999*; *Uetani et al., 2006*; *Wallace et al., 1999*), but are also highly expressed in mature neurons where they are thought to be presynaptic (*Um and Ko,*

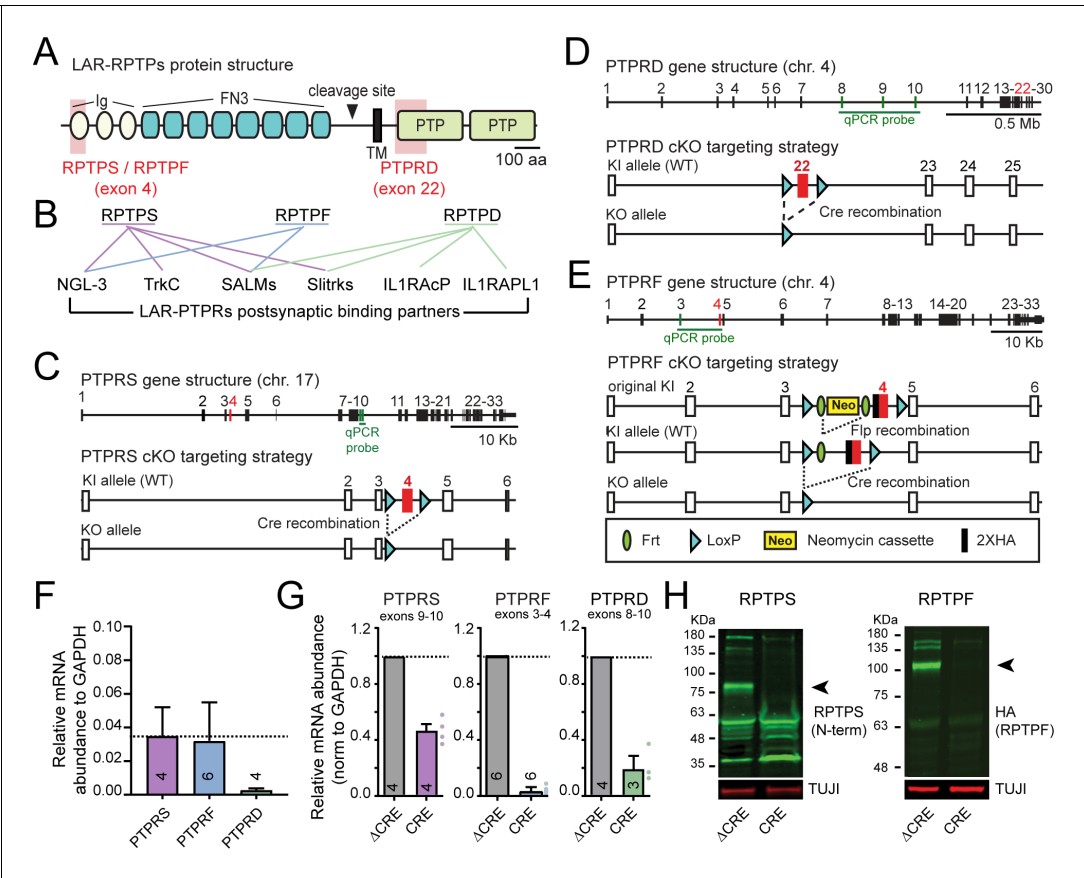

**Figure 1.** Targeting strategy and validation of LAR-PTPR conditional KO mice. (**A**) Schematic of LAR-RPTPs. Shaded red areas show regions targeted in the cKO mice (Ig, immunoglobulin domain; FN3, fibronectin type three repeat; TM, transmembrane region; PTP, phospho-tyrosine phosphatase domain). (**B**) Schematic of trans-synaptic interactions of LAR-RPTPs (*Um and Ko, 2013*). (**C–E**) Gene structure and targeting strategy for PTPRS (**C**) (*Bunin et al., 2015*), PTPRD (**D**) (*Farhy-Tselnicker et al., 2017*), and PTPRF cKO mice that were generated for this project (**E**). Note that the targeted exons are shown in red and the exons amplified by the RT-PCR probes used in *Figure 1F* are shown in green. (**F**) mRNA abundance (normalized to GAPDH) of PTPRS, PTPRF, and PTPRD as determined by RT-PCR in wildtype cultured hippocampal neurons. (**G**) Effect of Cre-dependent deletion of PTPRS, PTPRF and PTPRD on their mRNA level in hippocampal cultures from PTPRS, PTPRF and PTPRD cKO mice, validating the targeting strategy. Note that although the PTPRS cKO completely ablates RPTPS protein expression because deletion of exon 4 causes a frameshift in the mRNA, the PTPRS mRNA levels are reduced by 50% upon cre recombination, because non-sense mediated decay only partially decreases the mRNA levels for this gene. Quantitative data are means ± SEMs. (**H**) Immunoblot for PTPRS (left) and PTPRF (middle) validating the cKO strategy (black arrowheads indicate the specific bands for PTPRS and PTPRF ectodomains).

*2013*). Cell-adhesion interactions of LAR-RPTPs with numerous postsynaptic partners, including NGL-3, TrkC, SALMs, SliTrks, and IL1RAPs, suggest a major role for LAR-RPTPs in synapse formation (*Choi et al., 2016*; *Kwon et al., 2010*; *Li et al., 2015*; *Lin et al., 2018*; *Takahashi et al., 2011*; *Takahashi et al., 2012*; *Um et al., 2014*; *Woo et al., 2009*; *Yoshida et al., 2012*; *Yoshida et al., 2011*) (*Figure 1B*). Moreover, LAR-RPTP binding to postsynaptic partners is regulated by alternative splicing, indicating a dynamic complex (*Choi et al., 2016*; *Li et al., 2015*; *Yamagata et al., 2015a*; *Yamagata et al., 2015b*).

In *C. elegans*, the LAR-RPTP ortholog PTP3 is expressed in two isoforms that differ in their extracellular domains. The longer PTP-3A isoform is similar to vertebrate LAR-RPTPs, is specifically localized at synapses, and controls synapse morphology (*Ackley et al., 2005*). The shorter PTP-3B isoform lacks the Ig domains as well as the first four FN3 repeats, and functions in axon guidance (*Ackley et al., 2005*). Drosophila expresses two type IIA LAR-RPTPs, DLAR and Ptp69D, that have similar structures and are essential for axon guidance and for target recognition at neuromuscular junctions (NMJs) (*Desai et al., 1997*; *Krueger et al., 1996*) and at retinal photoreceptors (*Clandinin et al., 2001*; *Maurel-Zaffran et al., 2001*). Loss-of-function mutants of DLAR in

Drosophila exhibit smaller NMJs, larger active zones, and presynaptic release deficits (*Kaufmann et al., 2002*).

Thus, the results from studies of invertebrate LAR-RPTPs establish that they perform a major role in neuronal development, and additionally suggest a function in synapse formation and/or specification of synapse properties. However, the understanding of vertebrate LAR-RPTPs is less advanced. Elegant biochemical and structural studies demonstrated tight interactions of presynaptic LAR-RPTPs with a diverse set of postsynaptic adhesion molecules, including SliTrks, SALMs, TrkC, IL1-RAPs, and NGL3, suggesting that LAR-RPTPs function as synaptic adhesion molecules (*Karki et al., 2018*; *Kwon et al., 2010*; *Lie et al., 2016*; *Lin et al., 2018*; *Um et al., 2014*; *Woo et al., 2009*; *Yamagata et al., 2015a*; *Yamagata et al., 2015b*). This hypothesis is supported by extensive data demonstrating that the various postsynaptic interactors of LAR-RPTPs perform important synaptic functions (*Choi et al., 2016*; *Kwon et al., 2010*; *Li et al., 2015*; *Lie et al., 2016*; *Takahashi et al., 2012*; *Valnegri et al., 2011*; *Woo et al., 2009*; *Yim et al., 2013*; *Yoshida et al., 2012*). However, the data on the functions of LAR-RPTPs themselves are less clear. RNAi-mediated knockdowns of LAR-RPTPs in cultured neurons produced a massive loss of synapses and an impairment in neurotransmitter release (*Dunah et al., 2005*; *Han et al., 2018*; *Ko et al., 2015*) consistent with a key role for LAR-RPTPs in synapse formation, but constitutive LAR-RPTP KOs did not cause gross abnormalities in the overall brain structure or evidence of synapse loss (*Elchebly et al., 1999*; *Horn et al., 2012*). Constitutive PTPRD and PTPRS KO mice also exhibited reductions in body weight, postnatal lethality, and behavioral abnormalities that include ataxia, abnormal limb flexion, and memory deficits (*Kolkman et al., 2004*; *Meathrel et al., 2002*; *Uetani et al., 2000*; *Wallace et al., 1999*). However, these mice display only modest changes of synaptic transmission, such as a reduction in presynaptic release probability and in LTP (*Horn et al., 2012*; *Uetani et al., 2000*). At this point, no mice lacking all LAR-RPTPs have been analyzed, and it is thus possible that the relative lack of phenotypes observed for single LAR-RPTP KOs is due to functional redundancy among LAR-RPTPs. Moreover, no conditional deletions of LAR-RPTPs were examined at the synaptic levels, and developmental compensation could also have ameliorated an otherwise stronger phenotype.

To definitively examine the functions of LAR-RPTPs at synapses, we here characterized the effect of their deletions in hippocampal neurons using well-controlled genetic manipulations. We found that LAR-RPTPs do not perform an essential role in synapse formation, but are important for shaping the properties of excitatory synapses. Specifically, we demonstrate that LAR-RPTPs maintain the synaptic localization of NMDA- receptors (NMDARs) both in cultured neurons and in vivo, without affecting AMPA-receptor- (AMPAR-) mediated responses.

## Results

### Generation of single and triple LAR-RPTP conditional knockout mice

To better understand the role of LAR-RPTPs in synapse formation and/or specification of synapse properties, we obtained previously described conditional KO (cKO) mice for PTPRS (*Bunin et al., 2015*) and PTPRD (*Farhy-Tselnicker et al., 2017*) in which exons 4 and 22 were targeted, respectively (*Figure 1A,C,D*). In addition, we newly generated cKO mice for PTPRF in which we targeted exon 4 and introduced a N-terminal HA-epitope tag (*Figure 1A,E*). We validated the genetic strategy for all cKO mice by culturing hippocampal neurons from newborn mice, infecting the neurons at DIV4 with lentiviruses expressing an inactive mutant version of Cre recombinase (ΔCRE, used as a control) or active Cre recombinase (CRE; both Cre's are EGFP-tagged, contain a nuclear localization signal and are expressed under control of the ubiquitin promoter; see *Kaeser et al., 2011*). We then analyzed the neurons at DIV12.

Immunoblotting using an antibody to the ectodomain of PTPRS confirmed that PTPRS was undetectable in PTPRS cKO neurons after Cre expression (*Figure 1H*, left). Similarly, PTPRF protein, as monitored by immunoblotting with an anti-HA antibody, was absent in PTPRF cKO neurons expressing Cre (*Figure 1H*, right). Finally, RT-PCR demonstrated that PTPRD mRNAs were deleted in PTPRD cKO neurons after Cre expression (*Figure 1G*). Thus, the LAR-RPTP cKO mice we are using enable conditional deletion of all LAR-RPTPs as predicted.

## LAR-RPTPs are not essential for neuronal maturation or synapse formation

To delete all LAR-RPTP isoforms, we crossed PTPRS, PTPRF and PTPRD single cKO mice to generate triple LAR-RPTP cKO mice. We then investigated whether LAR-RPTPs control synaptogenesis by culturing hippocampal neurons from newborn PTPRS, PTPRF, and PTPRD single cKO mice as well as from triple LAR-RPTP cKO mice. We infected the neurons at DIV4 with lentiviruses encoding ΔCre (as a control) or Cre, and analyzed them at DIV12. We immunostained excitatory and inhibitory synapses using antibodies to vGluT1 and vGAT, respectively, and immunolabeled dendrites using MAP2 (*Figure 2A,E*). Strikingly, neither the single deletions of PTPRS, PTPRF, or PTPRD nor the triple deletion of all LAR-RPTPs altered the density of excitatory (*Figure 2A–B*) or inhibitory synapses (*Figure 2E–F*). Moreover, the size and staining intensity of vGluT1 and vGAT puncta were unchanged by LAR-RPTP deletions (*Figure 2C–D,G–H*). These data show that LAR-RPTPs, despite recruiting postsynaptic markers in artificial synapse formation assays (*Bomkamp et al., 2019*; *Woo et al., 2009*; *Yoshida et al., 2011*), are not required for synapse formation as such.

To examine whether the LAR-RPTP deletion alters neuronal development, we sparsely transfected cKO neurons expressing Cre or ΔCre with tdTomato, and analyzed their dendritic arborization, axon length, and soma size. However, deletion of all LAR-RPTPs in a neuron did not induced detectable changes in three basic morphological features (dendrites, axons, and cell body size), suggesting that LAR-RPTPs are not required for the basic development of neurons (*Figure 2I–J*).

## Presynaptic LAR-RPTPs regulate postsynaptic NMDARs

The lack of an essential role for LAR-RPTPs in the postmitotic development of neurons and/or in synapse formation is surprising, prompting us to assess a possible function of LAR-RPTPs in the elaboration of the electrical properties of neurons or in synaptic transmission.

Using patch-clamp recordings, we found that cultured neurons lacking individual LAR-RPTPs exhibit a normal capacitance and input resistance, consistent with a lack of effect on neuronal development (*Figure 4—figure supplement 1A–C*). Deletion of all LAR-RPTP isoforms induced a small reduction (~15%) in neuronal capacitance, but again did not affect the input resistance (*Figure 4—figure supplement 1D*).

Recordings of spontaneous mEPSCs or mIPSCs from control neurons and neurons lacking all LAR-RPTPs, obtained as described above, uncovered a significant decrease (~40%) in mEPSC but not in mIPSC frequency (*Figure 3A–B,D–E*). Neither the mEPSC nor the mIPSC amplitudes exhibited major changes (*Figure 3C,F*). We also observed a small but non-significant decrease in the amplitude of evoked AMPAR-mediated EPSCs (*Figure 4A*). Evoked NMDAR-mediated EPSCs, in contrast, exhibited a major decrease in amplitude (~40% decrease; *Figure 4B*). As a result of the decrease in NMDAR- but not AMPAR-EPSCs, the NMDA/AMPA ratio was also greatly decreased (~60%; *Figure 4C–D*). This selective impairment in postsynaptic NMDAR-mediated synaptic responses was not caused by a cell-autonomous postsynaptic function of LAR-RPTPs because postsynaptic deletions of LAR-RPTPs in sparsely transfected neurons had no effect on the NMDA/AMPA ratio (*Figure 4E–F*).

To test whether the selective decrease in NMDAR-EPSC amplitudes caused by the LAR-RPTP deletion might be due to a loss of total surface NMDARs, we quantified currents elicited in cultured hippocampal neurons by bath application of AMPA or NMDA (*Figure 4G–H*, *Figure 4—figure supplement 1E–F*). The current density of AMPAR-mediated responses was unchanged by the LAR-RPTP deletion (*Figure 4G*, *Figure 4—figure supplement 1E*). Strikingly, the current density of NMDAR-mediated responses was significantly enhanced, and not decreased (*Figure 4H*, *Figure 4—figure supplement 1F*). This result indicates that the decrease in evoked NMDAR-EPSCs is not due to a loss of surface NMDARs, but is caused by a loss of specifically synaptic NMDARs. To probe for a potential change in NMDAR protein levels or other components of synapses, we additionally analyzed the proteome of control and LAR-RPTP-deficient hippocampal neurons by quantitative immunoblotting (*Figure 5*). We detected no major changes in any protein examined, including NMDAR subunits, active zone proteins, or components of the neurotransmitter release machinery (*Figure 5*).

We also explored the possibility that the LAR-RPTP deletion might have changed the NMDAR composition, and quantified the relative inhibition of NMDAR-EPSCs by Ifenprodil, a selective GluN2A inhibitor. However, we detected no difference in Ifenprodil sensitivity between control

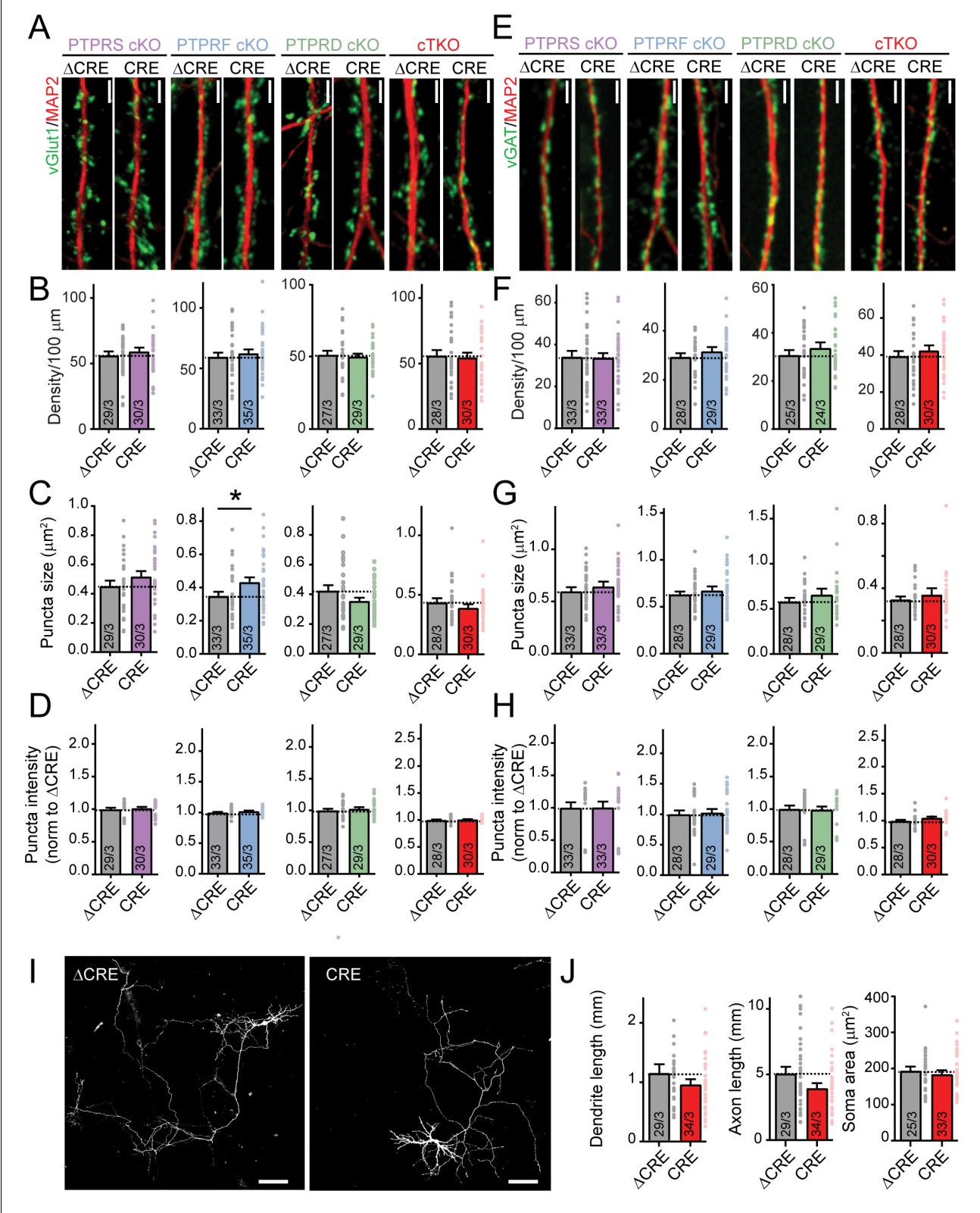

**Figure 2.** Deletion of LAR-RPTPs does not decrease synapse numbers or affect neuronal development in cultured hippocampal neurons. All experiments were performed with hippocampal neurons cultured from single PTPRS, PTPRF, or PTPRD cKO mice or triple LAR-RPTP cKO mice. Neurons were infected with lentiviruses expressing Cre recombinase (CRE) or a non-functional mutant version of Cre recombinase (ΔCRE, used as control). (**A**) Representative images of dendrites stained for an excitatory synaptic marker (vGlut1, green; MAP2 in red was used to visualize dendrites).
*Figure 2 continued on next page*

*Figure 2 continued*

Scale bar, 10 μm. (B–D) Quantifications of excitatory synapse density (B), puncta size (C) and puncta intensity (D). (E) Representative images of dendrites stained for an inhibitory synaptic marker (vGAT, green; MAP2 in red was used to visualize dendrites). Scale bar, 10 μm. (F–H) Quantifications of inhibitory synapse density (F), puncta size (G) and puncta intensity (H). (I) Representative images of individually transfected hippocampal neurons from LAR-RPTP triple cKO mice. Scale bar, 100 μm. (J) Summary graphs of dendrite, axon, and soma sizes as a function of the LAR-RPTP triple deletion. All data are means ± SEMs. Data comparing two conditions were analyzed by two-tailed unpaired Student's t-test (for B, PTPRS p=0.4667, ΔCRE n = 29/3, CRE n = 30/3; PTPRF p=0.5335, ΔCRE n = 33/3, CRE n = 35/3; PTPRD p=0.7261, ΔCRE n = 27/3, CRE n = 29/3; LAR-RPTP triple cKO mice p=0.8057, ΔCRE n = 28/3, CRE n = 30/3; for C, PTPRS p=0.2187, ΔCRE n = 29/3, CRE n = 30/3; for PTPRF *p=0.0253, ΔCRE n = 33/3, CRE n = 35/3; for PTPRD p=0.0958, ΔCRE n = 27/3, CRE n = 29/3; LAR-RPTP triple cKO mice p=0.3063, ΔCRE n = 28/3, CRE n = 30/3; for D, PTPRS p=0.6450, ΔCRE n = 29/3, CRE n = 30/3; PTPRF p=0.1505, ΔCRE n = 33/3, CRE n = 35/3; PTPRD p=0.4879, CRE n = 27/3, CRE n = 29/3; LAR-RPTP triple cKO mice p=0.4885, ΔCRE n = 28/3, CRE n = 30/3; for F, PTPRS p=0.8812, ΔCRE n = 33/3, CRE n = 33/3; PTPRF p=0.3262, ΔCRE n = 28/3, CRE n = 29/3; PTPRD p=0.3471, ΔCRE n = 25/3, CRE n = 24/3; LAR-RPTP triple cKO mice p=0.4300, ΔCRE n = 28/3, CRE n = 30/3; for G, PTPRS p=0.3232, ΔCRE n = 33/3, CRE n = 33/3; PTPRF p=0.1396, ΔCRE n = 28/3, CRE n = 29/3; PTPRD p=0.3039, CRE n = 25/3, CRE n = 24/3; LAR-RPTP triple cKO mice p=0.3876, ΔCRE n = 28/3, CRE n = 30/3; for H, PTPRS p=0.8706, ΔCRE n = 33/3, CRE n = 33/3; PTPRF p=0.7720, ΔCRE n = 28/3, CRE n = 29/3; PTPRD p=0.8650, ΔCRE n = 25/3, CRE n = 24/3, LAR-RPTP triple cKO mice p=0.0824, ΔCRE n = 28/3, CRE n = 30/3; for J, ΔCRE n = 29/3, CRE n = 34/3, p=0.5197 for dendrite length, p=0.0780 for axon length, ΔCRE n = 25/3, CRE n = 33/3, p=0.5488 for soma area).

(ΔCRE) or LAR-RPTP KO (CRE) neurons (*Figure 4I–K*). Overall, the electrophysiology and immuno-blotting results support the morphological finding that LAR-RPTPs are not required for synapse formation and maintenance (*Figure 2*), and reveal that LAR-RPTPs are essential for the maintenance of synaptic NMDARs via a trans-synaptic mechanism. In addition, the deletion of LAR-PTPRs may have had a modest effect on the release probability as judged by the decrease in mEPSC frequency, but this effect was small since it did not cause an impairment in AMPAR-mediated EPSCs.

## LAR-RPTP deletion in vivo impairs NMDAR-EPSCs monitored in CA1 neurons

Our results so far show that in cultured hippocampal neurons, LAR-RPTPs are not essential for synapse formation, but are required to control postsynaptic NMDAR. However, analyses of cultured neurons can misidentify or overlook major functions of a gene because of the necessarily non-physiological nature of neuronal cultures. To further explore the functions of LAR-RPTPs, we conditionally deleted all LAR-RPTPs in vivo. LAR-RPTPs are expressed at high levels in the hippocampus in both CA3 and CA1 pyramidal cells not only during development, but also in adult mice (*Figure 6—figure supplement 1A*) (*Saunders et al., 2018*). To study the role of LAR-RPTPs in vivo, we performed bilateral stereotactic injections of AAVs expressing inactive (ΔCre, control) or active Cre-recombinase into the CA3 region of newborn LAR-RPTP triple cKO mice (*Wu et al., 2019*). We then analyzed the properties of CA3 to CA1 Schaffer-collateral synapses in acute hippocampal slices at P30-37, using whole-cell patch-clamp recordings from CA1 pyramidal neurons (*Figure 6A*).

Measurements of spontaneous sEPSCs (*Figure 6—figure supplement 1B–C*) and of unitary synaptic AMPAR-mediated events using Sr$^{2+}$ in the extracellular medium (*Figure 6B*) showed that the presynaptic LAR-RPTP deletion caused a small but significant decrease in the frequency of sEPSCs, but had little effect on the amplitude of sEPSCs or of unitary EPSCs that reflects the size of AMPAR-mediated synaptic responses. Moreover, input/output measurements as a function of electrical Schaffer-collateral stimulation demonstrated that the LAR-RPTP deletion caused no change in AMPAR-mediated synaptic strength or synaptic connectivity (*Figure 6C*), consistent with the lack of an effect of the LAR-RPTP deletion on AMPARs or synapse numbers observed in cultured neurons. However, the presynaptic deletion of LAR-RPTPs in CA3 neurons led to a significant reduction in the NMDAR/AMPAR ratio in CA1 neurons (*Figure 6D*), similar to what we observed in hippocampal culture neurons (*Figure 4C–D*). Furthermore, we detected a modest increase in the coefficient of variation (CV) of NMDAR-EPSCs, but not of AMPAR-EPSCs (*Figure 6E*). Viewed together, these data indicate that deletion of all LAR-RPTPs in postmitotic neurons in vivo prior to synapse formation did not impede synaptogenesis, but caused a decrease in NMDAR-mediated synaptic responses without affecting AMPAR-mediated synaptic responses.

The relative reduction in the NMDAR/AMPAR ratio in vivo (*Figure 6D*) was smaller than that observed in cultured neurons (*Figure 4C–D*). This difference might be due to incomplete targeting of CA3 inputs in vivo, prompting us to employ an optogenetic approach to eliciting synaptic responses. To circumvent the problem of incomplete infection of CA3 region neurons, we bilaterally

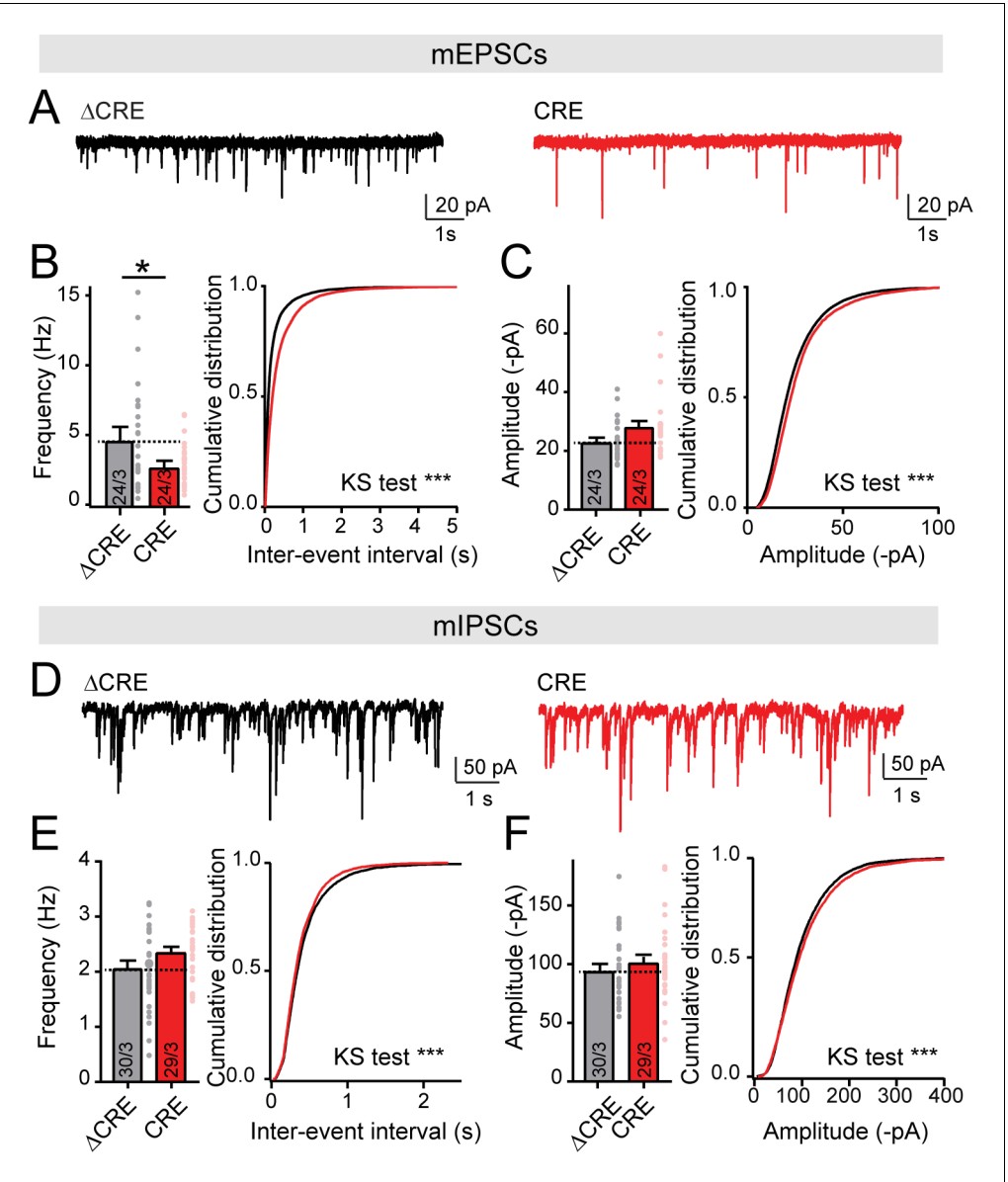

**Figure 3.** Deletion of LAR-RPTPs decreases mEPSCs frequency, but does not change mIPSCs frequency nor amplitude. All experiments were performed with hippocampal neurons cultured from triple LAR-RPTP cKO mice. Neurons were infected with lentiviruses expressing Cre recombinase (CRE) or a non-functional mutant version of Cre recombinase (ΔCRE, used as control). (**A–C**) mEPSCs recordings in hippocampal cultures from LAR-PTPR triple cKO mice (A, representative traces; B, summary graph and cumulative plot for frequency; C, summary graph and cumulative plot for amplitude). (**D–F**) Same as A-C for mIPSCs (D, representative traces; E, summary graph and cumulative plot for frequency; F, summary graph and cumulative plot for amplitude). All data are means ± SEMs. Data comparing two conditions were analyzed by two-tailed unpaired Student's t-test (for B, ΔCRE n = 24/3, CRE n = 24/3, *p=0.0384; for C, ΔCRE n = 24/3, CRE n = 24/3, p=0.0581; for E, ΔCRE n = 30/3, CRE n = 29/3, p=0.0697; for F, ΔCRE n = 30/3, CRE n = 29/3, p=0.3837). For cumulative plots statistical comparisons were performed using Kolmogorov-Smirnov (KS) tests (for B, ***p<0.0001; for C, ***p<0.0001; for E, ***p=0.0001; for F, ***p=0.0005).

co-injected newborn LAR-RPTP cKO mice with two AAVs. The first AAV encoded a Cre-dependent channelrhodopsin (ChiEF) (*Lin et al., 2009*) that was fused to tdTomato for visualization (AAV-DIO-ChiEF-tdTomato), while the second AAV encoded EGFP-fused Cre-recombinase under the control of the synapsin promoter (*Figure 7A*). As control, we used age and sex-matched BL6 mice, injected with the same set of AAVs. We sacrificed injected mice at P30 and performed whole-cell recordings

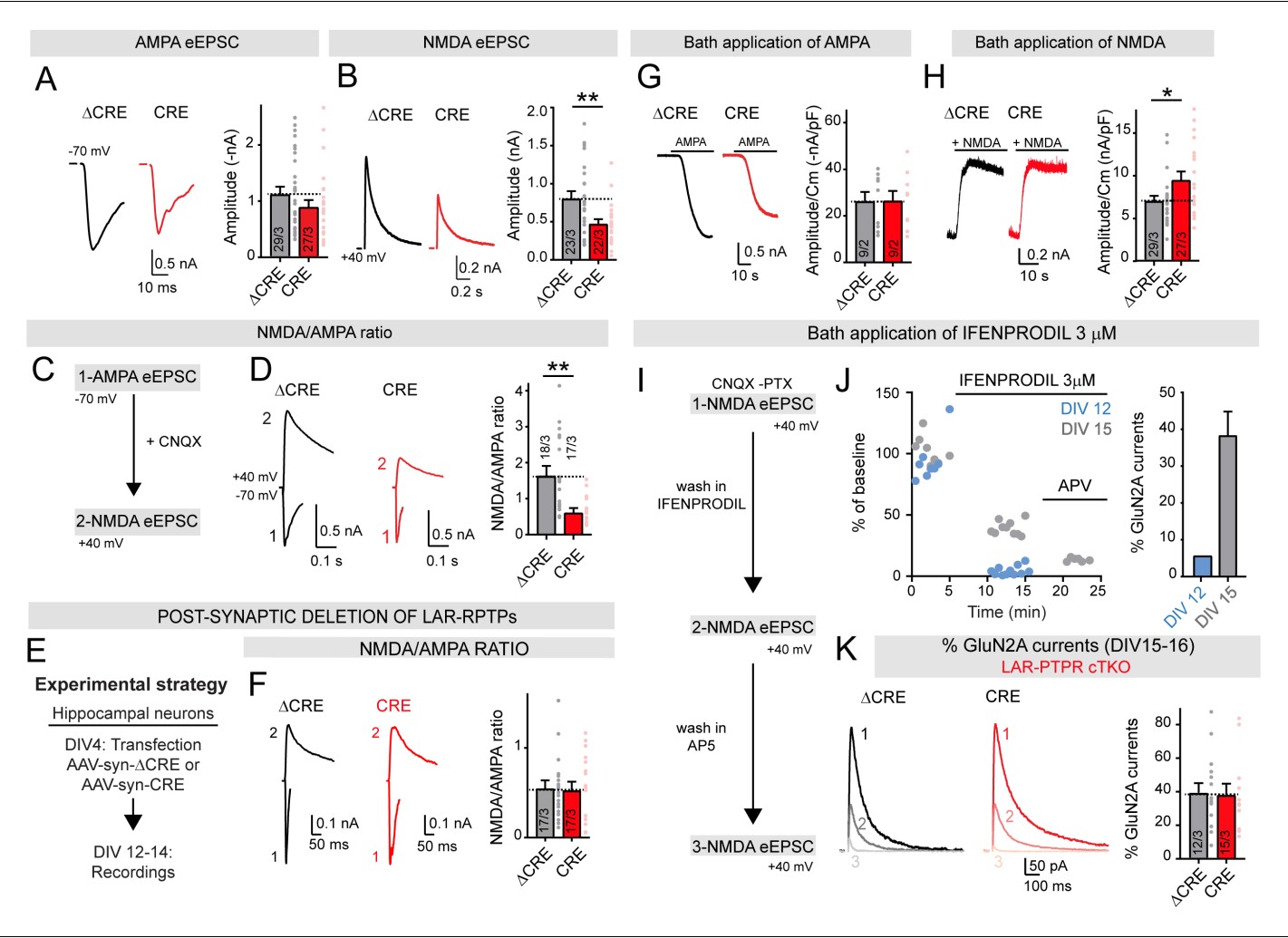

**Figure 4.** Deletion of LAR-RPTPs suppresses synaptic NMDAR- but not AMPAR-mediated responses in cultured hippocampal neurons. All experiments were performed with hippocampal neurons cultured from triple LAR-RPTP cKO mice. Neurons were infected with lentiviruses expressing Cre recombinase (CRE) or a non-functional mutant version of Cre recombinase (ΔCRE, used as control). (**A**) AMPAR-EPSC amplitudes recorded at −70 mV in presence of PTX are not significantly altered by deletion of all three LAR-RPTPs (left, representative traces; right, summary graphs). (**B**) NMDAR-EPSC amplitudes recorded at +40 mV in presence of PTX and CNQX are decreased by deletion of all three LAR-RPTPs (left, representative traces; right, summary graphs). (**C**) Experimental strategy for recording the NMDA/AMPA ratio in hippocampal cultures from triple LAR-RPTP cKO mice infected with ΔCRE (WT) or CRE (KO). (**D**) NMDAR/AMPAR-EPSC ratios are suppressed by deletion of LAR-RPTPs. AMPAR-EPSCs were recorded at −70 mV in the presence of PTX, and NMDAR-EPSCs were then recorded at +40 mV in the presence of CNQX (left, representative traces; right, summary graphs). (**E**) Experimental strategy for post-synaptic deletion of LAR-RPTPs in hippocampal cultures. Cultures were sparsely transfected with AAV-synapsin-ΔCRE-EGFP or AAV-synapsin-CRE plasmids at DIV4. Transfected cells expressing EGFP were patched at DIV12-14. (**F**) NMDA/AMPA ratio in EGFP positive neurons, expressing Cre or ΔCre recombinase, showing that post-synaptic deletion of LAR-RPTPs does not impair AMPAR- or NMDAR-mediated EPSCs. (**G**) AMPAR-responses elicited by bath-applied AMPA (1 μM) are unchanged by deletion of LAR-RPTPs (left, representative traces; right, summary graphs of peak current densities). (**H**) NMDAR-responses elicited by bath-applied NMDA (10 μM) and glycine (10 μM) are increased by deletion of LAR-RPTPs (left, representative traces; right, summary graphs of peak current densities). (**I–K**) Effect of a bath application of 3 μM Ifenprodil, a specific GluN2B blocker, on control hippocampal cultures at different maturation stages (DIV12, blue or DIV15, grey). (**I**) Schematic of experimental procedures: whole-cell recordings of hippocampal neurons were performed in the presence of CNQX and PTX in the bath to isolate NMDAR-EPSCs that were recorded at +40 mV. Ifenprodil (3 μM) and AP5 were sequentially added to the bath and NMDAR-EPSCs were recorded. (**J**) Sample traces of NMDAR-EPSCs (% of baseline) as a function of time. Application of Ifenprodil completely blocked NMDAR currents at DIV12 (in blue), but only partially reduced NMDAR currents at DIV15 (in grey), confirming that with maturation neurons switched from GluN2B only containing NMDARs to GluN2A containing NMDARs. Subsequent application of AP5 completely inhibited NMDAR-EPSCs at DIV15; right summary graph depicts the percentage of GluN2A currents at DIV 12 and DIV15. (**K**) Reduction of NMDAR-EPSCs induced by Ifenprodil was unchanged upon deletion of LAR-RPTPs, suggesting that the NMDAR composition does not depend on LAR-RPTPs. All data are means ± SEMs. Data comparing two conditions were analyzed by two-tailed unpaired Student's t-test (for A, ΔCRE n = 29/3, CRE n = 27/3, p=0.1993; for B, ΔCRE n = 23/3, CRE n = 22/3, **p=0.0057; for D, ΔCRE n = 18/3,

*Figure 4 continued on next page*

Figure 4 continued

CRE n = 17/3, **p=0.0010; for F, ΔCRE n = 17/3, CRE n = 17/3, p=0.8726; for G, ΔCRE n = 9/2, CRE n = 9/2, p=0.9682; for H, ΔCRE n = 25/3, CRE n = 20/3, *p=0.0212, for K, ΔCRE n = 12/3, CRE n = 15/3, p=0.8891).

The online version of this article includes the following figure supplement(s) for figure 4:

**Figure supplement 1.** Triple deletion of LAR-RPTPs causes a modest reduction in the cell capacitance but not in the input resistance of cultured hippocampal neurons.

from CA1 pyramidal neurons in acute hippocampal slices in response to optical stimulation of CA3 region neurons (*Figure 7A*). The presynaptic deletion of LAR-RPTP caused a large reduction in the NMDAR/AMPAR ratio (*Figure 7B*), confirming the results obtained with electrical stimulation (*Figure 6D*).

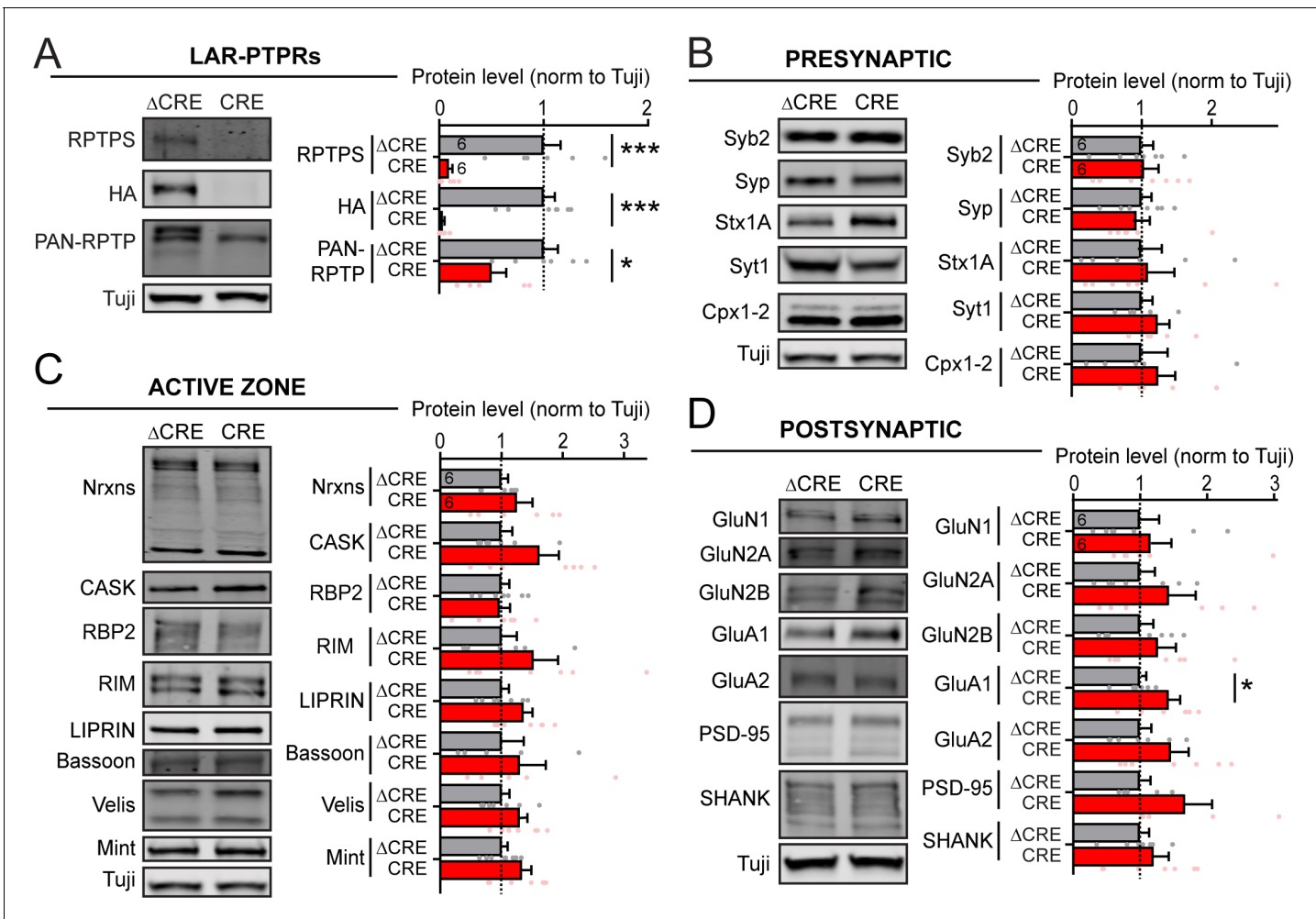

**Figure 5.** Global deletion of LAR-RPTPs does not significantly change the levels of pre- or postsynaptic proteins, including those of NMDARs and AMPARs. All experiments were performed with hippocampal neurons cultured from triple LAR-RPTP cKO mice. Neurons were infected with lentiviruses expressing Cre recombinase (CRE) or a non-functional mutant version of Cre recombinase (ΔCRE, used as control). (A–D) Representative images of immunoblots (left in each panel) and summary graphs of protein levels (normalized for controls) obtained in hippocampal neurons cultured from LAR-RPTP triple cKO mice. Neurons were infected with lentiviruses expressing Cre or ΔCre at DIV4 as described above, and analyzed at DIV12-14. Proteins are organized into groups comprising LAR-RPTPs (A) and marker proteins for presynaptic (B), active zone (C), and postsynaptic (D) specializations. Graphs show means ± SEM. Statistical significance was assessed with the two-tailed unpaired Student's t-test (ΔCRE n = 6, CRE n = 6, *p<0.05, ***p<0.001).

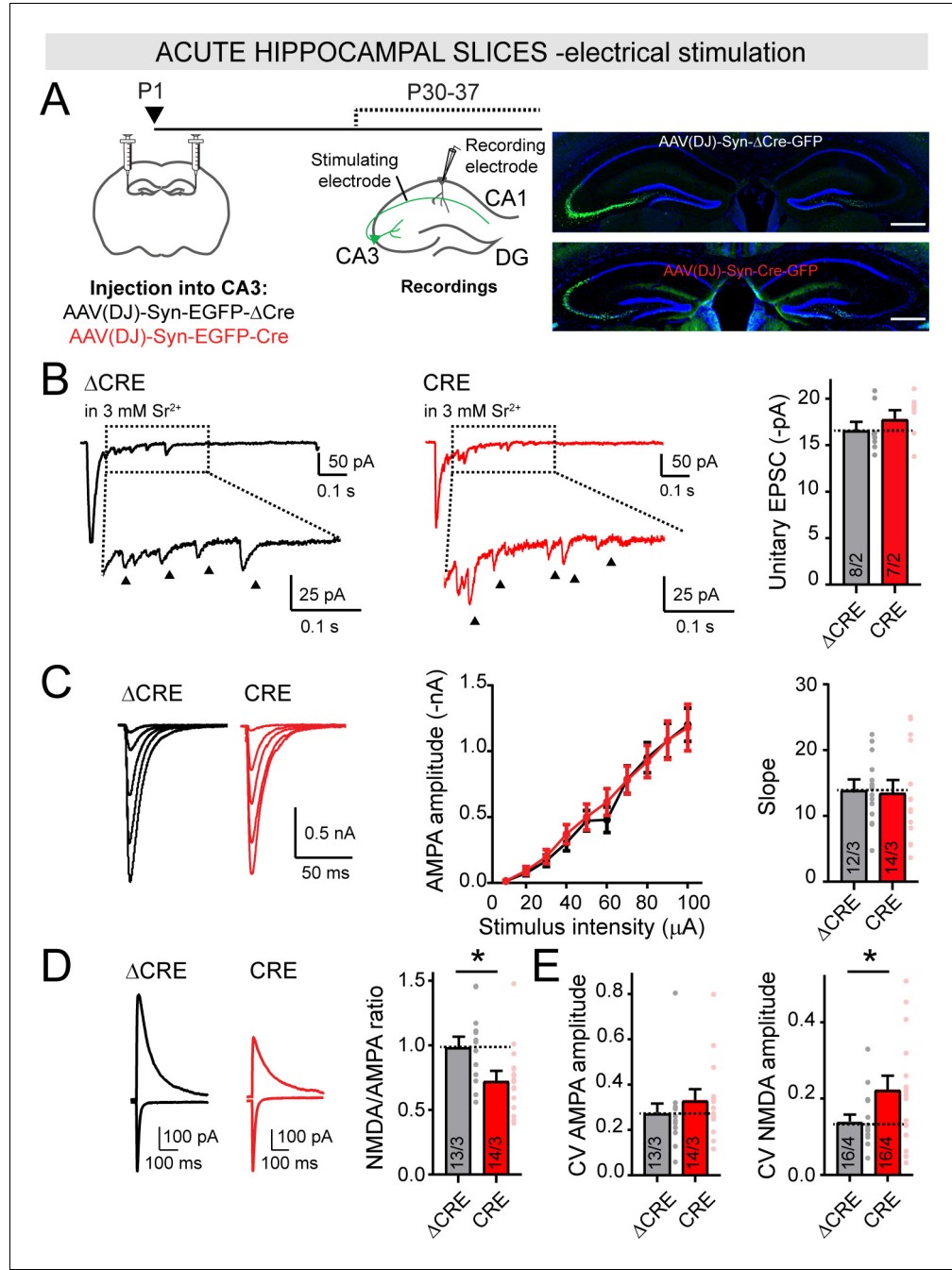

**Figure 6.** Deletion of LAR-RPTPs in vivo selectively decreases the NMDAR-/AMPAR-EPSC ratio at Schaffer collateral synapses in the hippocampus without affecting AMPAR-mediated synaptic connectivity. (**A**) Schematic of experimental strategy for panels B-E (left), and representative images of injection sites in the CA3 region of the hippocampus (right; scale bar = 0.5 mm). The CA3 region of newborn littermate LAR-RPTP triple cKO mice was stereotactically infected with AAVs expressing ΔCre (as a control) or Cre, and Schaffer-collateral EPSCs elicited by electrical stimulation were recorded from CA1 neurons at P30-37. (**B**) AMPAR-mediated unitary EPSCs elicited by electrical stimulation of Schaffer collaterals in the presence of 3 mM $Sr^{2+}$ extracellularly (0 $Ca^{2+}$) are not altered by the deletion of LAR-RPTPs. EPSC peak amplitude (showed by arrows) was analyzed within 50–500 ms after the stimulation (dotted box). Results confirm that removal of LAR-RPTPs does not affect post-synaptic AMPAR amplitudes. (**C**) Input-output curves of AMPAR-EPSCs confirm that AMPAR-mediated synaptic connectivity is not affected by deletion of LAR-RPTPs from CA3 neurons. (**D**) Presynaptic LAR-RPTP deletion suppresses the NMDA/AMPA ratio (left, representative traces; right, summary graph). The peak amplitudes of AMPAR-EPSCs were recorded at −70 mV in PTX, and composite AMPAR- and NMDAR-EPSCs were then recorded at +40 mV in the

*Figure 6 continued on next page*

*Figure 6 continued*

same cells, with the NMDAR-EPSC component quantified at 50 ms after the stimulus. (**E**) Presynaptic LAR-RPTP deletion has no effect on the coefficient of variation (CV) of AMPAR-EPSCs (left) but cause a modest increase in the CV of NMDAR-EPSCs (right). All data are means ± SEMs. Data comparing two conditions were analyzed by two-tailed unpaired Student's t-test (for B, ΔCRE n = 8/2, CRE n = 7/2, p=0.3711; for C, ΔCRE n = 12/3, CRE n = 14/3, p=0.8679 for slope; for D, ΔCRE n = 13/3, CRE n = 14/3, *p=0.0246 for NMDA/AMPA ratio; for E, ΔCRE n = 13/3, CRE n = 14/3, p=0.3606 for AMPAR-EPSC CV and ΔCRE n = 16/4, CRE n = 16/4, *p=0.0439 for NMDAR-EPSC CV). Input-output curves were analyzed by two-way ANOVA for repetitive measurements followed by Bonferroni post-hoc test (for C, ΔCRE n = 12/3, CRE n = 14/3, interaction p=0.9833).

The online version of this article includes the following figure supplement(s) for figure 6:

**Figure supplement 1.** LAR-RPTPs are highly expressed in hippocampal pyramidal neurons (**A**), and in vivo deletion of LAR-RPTPs does not significantly affect the frequency or amplitude of spontaneous EPSCs monitored in acute slices (**B–C**).

## Discussion

Synaptic cell-adhesion molecules control synapse formation and synapse specification throughout life, and thereby shape the organization and properties of neural circuits (*Südhof, 2018*). LAR-RPTPs

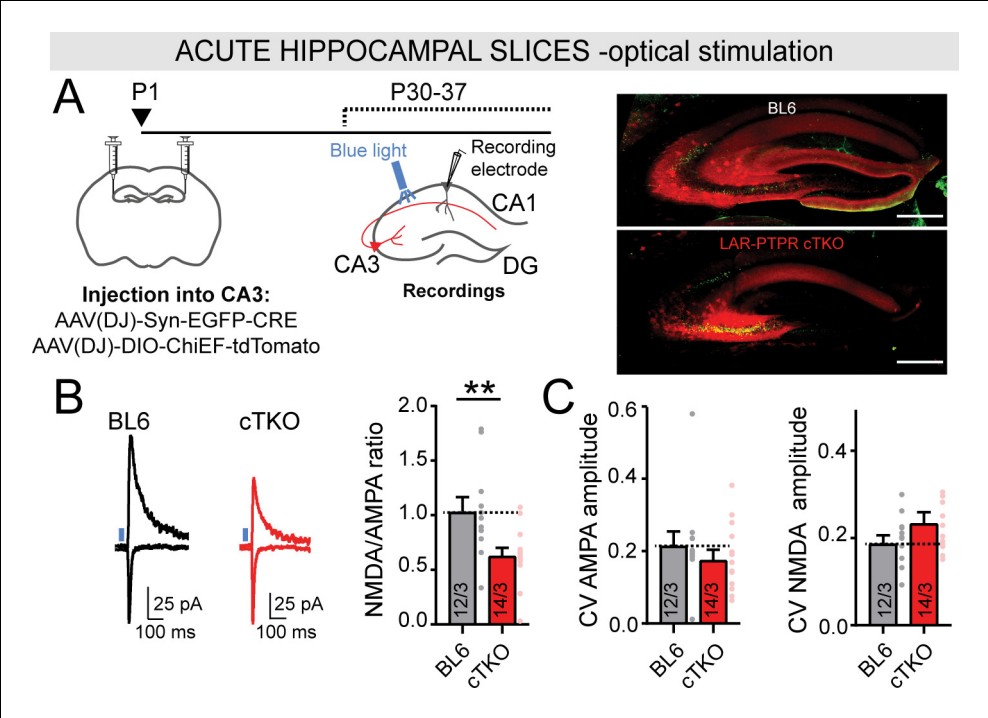

**Figure 7.** Optical stimulation of Schaffer collaterals from LAR-PTPR KO mice confirmed a loss of synaptic NMDA receptors, resulting in a decrease of the NMDAR-/AMPAR-EPSC ratio. (**A**) Schematic of experimental strategy for panels B-C (left), and representative images of infected slices expressing nuclear EGFP-tagged Cre (green) and tdTomato-tagged ChiEF (red)(right; scale bar = 0.2 mm). The CA3 regions of newborn LAR-RPTP triple cKO mice and unrelated WT mice were infected with two AAVs encoding a. Cre, and b. Cre-dependent ChieF, and Schaffer-collateral EPSCs elicited by optical stimulation were recorded from CA1 neurons at P30-37. (**B**) Presynaptic LAR-RPTP deletion suppresses the NMDAR-/AMPAR-EPSC ratio (left, representative traces; right, summary graph). Experiments were performed as for *Figure 6D*, except that optical stimulation was used. (**C**) Presynaptic LAR-RPTP deletion has no effect on the coefficient of variation (CV) of AMPAR-EPSCs (left) or NMDAR-EPSCs (right) measured in response to optical stimulation. All data are means ± SEMs. Data comparing two conditions were analyzed by two-tailed unpaired Student's t-test (for B, WT BL6 mice n = 12/3, LAR-RPTP cKO mice n = 14/3, **p=0.0059; for C, WT BL6 n = 12/3, LAR-RPTP cKO mice n = 14/3, p=0.3817 for AMPAR CV, and p=0.1361 for NMDAR CV).

are abundant presynaptic cell-adhesion molecules that are thought to be major drivers of synapse formation (reviewed in *Takahashi and Craig, 2013*; *Um and Ko, 2013*). LAR-RPTPs interact with a large number of key postsynaptic adhesion molecules, and these interactions are thought to mediate the synaptogenic function of LAR-RPTPs (reviewed in *Won and Kim, 2018*). Moreover, in Drosophila and *C. elegans* LAR-RPTPs are important regulators of synapse morphology and target specificity, consistent with a function in synapse formation (*Ackley et al., 2005*; *Clandinin et al., 2001*; *Kaufmann et al., 2002*). However, in vertebrates the synaptic functions of LAR-RPTPs are less clear because different approaches have yielded distinct results, and because the synaptic phenotypes of conditional deletions of LAR-RPTPs, arguably the most rigorous approach to examining their functions, have not been explored.

To address this important question, we have now generated single and triple conditional LAR-RPTP KO mice, and studied the role of LAR-RPTPs in synapse formation and synaptic transmission in cultured neurons and in vivo. This approach enabled us to avoid three potential difficulties associated with studies of the synaptic functions of LAR-RPTPs by other approaches. First, by using conditional genetic deletions in neurons after neurogenesis but before synapse formation, we eliminated the effects of LAR-RPTP deletions on earlier developmental stages in which LAR-RPTPs are known to have major roles (*Chagnon et al., 2004*; *Meathrel et al., 2002*; *Uetani et al., 2006*; *Wallace et al., 1999*). Second, by targeting all LAR-RPTP genes, we ruled out the possibility of overlooking phenotypes that may have been occluded by redundancy among the three LAR-RPTP genes. Third, by studying manipulations both in cultured neurons and in vivo, we avoided potential culture artifacts, but at the same time were able to examine molecular mechanisms more precisely using culture conditions.

Surprisingly, our results demonstrate that LAR-RPTPs do not perform an essential role in synapse formation as such in vertebrate neurons. LAR-RPTPs were not required for establishing or maintaining synaptic connections in cultured neurons (*Figures 2* and *3*) or in vivo in a well-defined hippocampal circuit (*Figures 6* and *7*). Specifically, single deletion of individual LAR-RPTPs or global deletion of all LAR-RPTPs did not change the number of excitatory or inhibitory synapses, did not induce alterations in axonal outgrowth or dendritic branching, and did not affect the synaptic connectivity of hippocampal Schaffer collaterals as measured by AMPAR-mediated input/output curves. Thus, similar to neurexins (*Südhof, 2017*) but different from latrophilins (*Sando et al., 2019*), LAR-PTPRs are not required for the formation or maintenance of synapses in hippocampal neurons. Naturally these results are not at odds with a role for LAR-RPTPs in axon guidance and other developmental processes, a role that has been well established (*Coles et al., 2011*; *Desai et al., 1997*; *Garrity et al., 1999*; *Krueger et al., 1996*; *Nakamura et al., 2017*; *Sun et al., 2000*; *Uetani et al., 2006*) and would not have become manifest in our experimental design.

However, our results demonstrate that LAR-RPTPs do perform an important function in shaping synapse properties in mature neurons by controlling postsynaptic NMDAR-mediated responses. This function is consistent with the continued high-level expression of LAR-RPTPs in mature neurons (*Figure 6—figure supplement 1A*). In cultured neurons, we showed that the LAR-RPTP deletion caused a decrease in NMDAR-EPSCs that was due to a relative loss of NMDARs from postsynaptic sites and not to a decrease in NMDAR proteins levels. We observed a large decrease in synaptic NMDAR-EPSCs evoked by action potentials, but a significant increase in NMDAR-responses elicited by direct NMDA application (*Figure 4*). The decrease in synaptic NMDAR-EPSCs was likely due to a presynaptic mechanism since the postsynaptic LAR-RPTP deletion had no effect on NMDAR-EPSCs (*Figure 4*). The NMDAR phenotype was confirmed in vivo, where we demonstrated that the presynaptic LAR-RPTP deletion did not impair postsynaptic AMPAR-mediated synaptic responses, but decreased the ratio of NMDAR- to AMPAR-EPSCs (*Figures 6* and *7*). The LAR-PTPR function in regulating NMDARs resembles that of neurexin-1, suggesting a possible mechanistic convergence (*Dai et al., 2019*).

Our findings suggest that at least in hippocampal CA3-CA1 synapses, the current hypotheses about LAR-RPTP functions need to be revised. These results indicate that the well-described interactions of LAR-RPTPs with various postsynaptic adhesion molecules (*Choi et al., 2016*; *Kwon et al., 2010*; *Li et al., 2015*; *Lin et al., 2018*; *Takahashi et al., 2011*; *Takahashi et al., 2012*; *Um et al., 2014*; *Woo et al., 2009*; *Yoshida et al., 2012*; *Yoshida et al., 2011*) may, at least in part, control the properties of NMDARs. Thus, LAR-RPTPs major function at the synapse appears to be to enable proper specification of synaptic properties, and not to mediate actual establishment of synapses and wiring of circuits. However, we would like to caution that naturally our study is also subject to several

limitations. Our analyses were restricted to synaptic connectivity and to AMPAR- and NMDAR-mediated synaptic responses, and we did not examine other synaptic properties such as various forms of plasticity that may or may not be impaired. It is possible that in addition to the robust effect of the LAR-RPTP deletions on NMDAR-EPSCs, the deletions also have an effect, albeit smaller, on release probability as suggested by their modest impact on mEPSC frequency, which would agree well with previous studies (*Horn et al., 2012*; *Uetani et al., 2000*). Moreover, we explored only one particular brain region (the hippocampus), and focused on only one particular synapse (Schaffer-collateral synapses). It is plausible that the deletion of LAR-RPTP could have a different phenotype in other brain regions and other synapses. Finally, we did not rule out potential redundancy of LAR-RPTPs with other unrelated cell-adhesion molecules, such as neurexins. Despite these limitations, however, our results indicate that in hippocampal synapses, LAR-RPTPs are important determinants of synapse properties that contribute to regulating NMDAR mediated responses at synapses and thereby shape the properties of neural circuits.

# Materials and methods

## Key resources table

| Reagent type (species) or resource | Designation | Source or reference | Identifiers | Additional information |
|---|---|---|---|---|
| Strain, strain background (*Mus musculus*) | PTPRS KO mice | UC Davis KOMP repository | RRID: IMSR_KOMP: CSD76529-1c-Mbp | |
| Strain, strain background (*Mus musculus*) | PTPRD | Welcome trust Sanger institute | RRID: IMSR_EM:11805 | |
| Strain, strain background (*Mus musculus*) | PTPRF | Südhof lab | | |
| Strain, strain background (*Mus musculus*) | C57BL/6J | The Jackson laboratory | RRID: IMSR_JAX:000664 | |
| Strain, strain background (*Mus musculus*) | ACTB-Flpe | The Jackson laboratory | RRID: IMSR_JAX:005703 | |
| Cell line (include species here) | HEK293T cells | ATCC | RRID: CVCL_0063 | |
| Antibody | Anti-PTPRS | Südhof lab | PAC9986, RRID: AB_2802087 | 1:500 |
| Antibody | Anti-HA | Biolegend | 901515, RRID: AB_2565334 | 1:1000 |
| Antibody | Anti-PAN-RPTP | Neuromab | 75–194, RRID: AB_2174700 | 1:500 |
| Antibody | Anti-Nrxns | Südhof lab | G394, RRID: AB_2800397 | 1:500 |
| Antibody | Anti-CASK | Neuromab | 75–000, RRID: AB_2068730 | 1:1000 |
| Antibody | Anti-RBP2 | Südhof lab | 4193, RRID: AB_2617050 | 1:1000 |
| Antibody | Anti-RIM | Südhof lab | U1565, RRID: AB_2617054 | 1:1000 |
| Antibody | Anti-liprin | Südhof lab | 4396, RRID: AB_2617056 | 1:1000 |
| Antibody | Anti-Bassoon | Synaptic Systems | 141021, RRID: AB_2066979 | 1:1000 |
| Antibody | Anti-Velis | Südhof lab | U049, RRID: AB_2802084 | 1:1000 |
| Antibody | Anti-Mint | Südhof lab | P932, RRID: AB_2802085 | 1:1000 |
| Antibody | Anti-Synaptobrevin 2 | Synaptic Systems | 104211, RRID: AB_887811 | 1:1000 |
| Antibody | Anti-Synaptophysin | Synaptic Systems | 101011, RRID: AB_887824 | 1:1000 |
| Antibody | Anti-Syntaxin | Synaptic Systems | HPC-1, RRID: AB_887843 | 1:1000 |
| Antibody | Anti-Synaptotagmin 1 | Synaptic Systems | 105011, RRID: AB_887832 | 1:1000 |
| Antibody | Anti-Complexin | Südhof lab | L669, RRID: AB_2802086 | 1:1000 |
| Antibody | Anti-GluN1 | Synaptic systems | 114 011, RRID: AB_887750 | 1:1000 |
| Antibody | Anti-GluN2A | Invitrogen | A6473RRID: AB_10376044 | 1:1000 |
| Antibody | Anti- GluN2B | Neuromab | 75–097, RRID: AB_10673405 | 1:1000 |
| Antibody | Anti-GluA1 | Synaptic Systems | 182003, RRID: AB_2113441 | 1:1000 |

*Continued on next page*

Continued

| Reagent type (species) or resource | Designation | Source or reference | Identifiers | Additional information |
|---|---|---|---|---|
| Antibody | Anti-GluA2 | Synaptic Systems | 182103, RRID: AB_2113732 | 1:1000 |
| Antibody | Anti-PSD-95 | Neuromab | 75–028, RRID: AB_2292909 | 1:1000 |
| Antibody | Anti-PAN-SHANK | Neuromab | 75–089, RRID: AB_10672418 | 1:1000 |
| Antibody | Anti-Tubulin | Sigma | T2200, RRID: AB_262133 | 1:5000 |
| Antibody | Anti-vGluT1 | Millipore | AB5905, RRID: AB_2301751 | 1:1000 |
| Antibody | Anti-vGAT | Millipore | AB5062P, RRID: AB_2301998 | 1:1000 |
| Antibody | Anti-MAP2 | Encor Biotech | CPCA-MAP2, RRID: AB_2138173 | 1:500 |
| Antibody | Alexa fluor 546, goat anti mouse IgG | Invitrogen | A-11003, RRID: AB_2534071 | 1:1000 |
| Antibody | Alexa fluor 488, goat anti guinea pig IgG | Invitrogen | A-11073 RRID: AB_2534117 | 1:1000 |
| Antibody | Alexa fluor 633, goat anti mouse IgG | Invitrogen | A-21050 RRID: AB_2535718 | 1:1000 |
| Antibody | IRDye 680LT donkey anti mouse | Licor | 926–68022, RRID: AB_621848 | 1:10000 |
| Antibody | IRDye 680LT donkey anti rabbit | Licor | 926–68023, RRID: AB_10706167 | 1:10000 |
| Antibody | IRDye 800CW donkey anti mouse | Licor | 926–32212, RRID: AB_10715072 | 1:10000 |
| Antibody | IRDye 800CW donkey anti rabbit | Licor | 926–32213, RRID: AB_621848 | 1:10000 |
| Chemical compound, drug | Picrotoxin | Tocris | 1128 | |
| Chemical compound, drug | CNQX | Tocris | 1045 | |
| Chemical compound, drug | AP5 | Tocris | 0106 | |
| Chemical compound, drug | Tetrodotoxin citrate | ARC | 0640 | |
| Chemical compound, drug | Ifenprodil | Tocris | 2892 | |
| Chemical compound, drug | CTZ | Tocris | 0713 | |
| Software, algorithm | Clampfit | Molecular device | N/A | |
| Software, algorithm | Igor Pro | Wavemetrics Inc | RRID:SCR_000325 | |
| Software, algorithm | FIJI | NIH | RRID:SCR_002285 | |
| Software, algorithm | pClamp | Molecular device | RRID:SCR_011323 | |
| Software, algorithm | Prism | Graphpad software inc | RRID: SCR_002798 | |
| Software, algorithm | Image Studio Lite | Licor | RRID: SCR_014211 | |
| Software, algorithm | NIS-elements | Nikon | RRID:SCR_002776 | |

## Animals

PTPRS cKO mice were purchased from the UC Davis KOMP repository (Ptprs_tm1c_D11, ES cell clone ID: DEPD00535_1_D11, RRID: IMSR_KOMP:CSD76529-1c-Mbp) (*Bunin et al., 2015*). PTPRD mice with cKO mice potential were obtained from the Welcome Trust Sanger Institute (Ptprdtm2a (KOMP)Wtsi, colony prefix MEXY, ESC clone ID: EPD0581_9_D04, RRID: IMSR_EM:11805) (*Farhy-Tselnicker et al., 2017*), and crossed to Flp mice to remove the Neo cassette (Jackson Laboratory, JAX:005703, RRID: IMSR_JAX:005703). PTPRF cKO mice were generated as previously described (*Sclip et al., 2016*), by flanking exon 4 with loxP sites. An HA-tag was introduced in front of the first

Ig domain in PTPRF mice to tag the protein (see *Figure 1A* and *Figure 1E* for the targeting strategy). PTPRF mice were also crossed to Flp mice to remove the Neo cassette. PTPRS, PTPRF and PTPRD mice were genotyped by PCR using the following standard program: 95°C 2', (94°C 30'', 60°C 30'', 72°C 1' x 35 cycles), 72°C 7'. The following oligonucleotide primers were used: AS15044 TTTCTGGCACTGCAGGGTTTCCCAAG, and AS15045 TCTGAATGGAGCACACCCTTAAGCCC for PTPRS single cKO (190 bp bands present in WT mice, 366 bp band in PTPRS cKO mice), AS15091 GGAGCTTGGAATAACCAGGA, AS15092 TACCATGCTACAGGTAGCAG and AS15093 CGG TAGAATTTCGACGACCT for PTPRF single cKO (297 bp bands present in WT mice, 369 bp band in PTPRF cKO mice), and AS15148 ATGTTTAGCTGGCCCAAATG, and AS15149 CGCTTCCTC GTGC TTTACGGTAT for PTPRD single cKO (379 bp bands present in WT mice, 526 bp band in PTPRD cKO mice). PTPRS, PTPRD and PTPRF single cKO mice were crossed together to homozygosity to obtain LAR-RPTP triple cKO mice. C57BL/6J (Jackson Laboratory, JAX:000664, RRID: IMSR_JAX: 000664) mice were purchased from the Jackson laboratory. Mice were group-housed on a 12 hr light-dark cycle with access to food and water ad libitum. Animal experiments were conducted following protocols approved by the Administrative Panel on Laboratory Animal Care at Stanford University (Protocol number APLAC-20787).

## Plasmids

The following plasmids were used: lentiviral vectors expressing EGFP tagged Cre recombinase (FUW-NL-EGFP-CRE) or EGFP tagged ΔCre recombinase (FUW-NL-EGFP-ΔCRE) under the Ubiquitin promoter, lentivirus helper plasmids (VSVG expression vector, pRRE and pRSV-REV), lentiviral vectors expressing tdTomato under the synapsin promoter (FSW-tdTomato), pAAV-Syn-EGFP-CRE and pAAV-Syn-EGFP-ΔCRE, AAV-DJ helper plasmids (pHelper and pRC-DJ), pAAV-CAG-DIO-ChiEF-tdTomato.

## Lentivirus preparation

Lentiviruses expressing EGFP-CRE and EGFP-ΔCRE under the control of the Ubiquitin promoter were produced as previously described (*Kaeser et al., 2011*). Briefly, FUW-NL-EGFP-CRE or FUW-NL-EGFP-ΔCRE plasmids were co-transfected with helper plasmids (VSVG expression vector, pRRE and pRSV-REV) in HEK293T cells using calcium phosphate. After 48 hr, cell media containing the lentiviruses were collected, snap-frozen in liquid N2, and stored at −80 C.

## AAV preparation

AAVs (serotype DJ) expressing Synapsin-EGFP-CRE and Synapsin-EGFP-ΔCRE, CAG-DIO-ChiEF-tdTomato were prepared by co-transfecting them into HEK293T cells with pHelper and pRC-DJ plasmids. Transfected cells were collected 72 hr later, lysed, and loaded into a iodixanol gradient. The 40% iodixanol fraction containing the virus was harvested, washed and concentrated with a 100,000 MWCO filter. AAVs were stored at −80 C before use.

## Neuronal cultures and transfection

Neuronal cultures were prepared from PTPRS, PTPRF, PTPRD single cKO mice or LAR-RPTP triple cKO pups at P0. Both female and male mice were used. Briefly, hippocampi were dissected and digested with papain, plated on Matrigel coated coverslip in 24-well plates. Cells were cultured in Neurobasal media supplemented with B27 supplement, Glutamax and 2 mM AraC for 12–16 days. For neuronal morphology studies (*Figure 2I–J*), neurons were sparsely transfected at DIV9 with synapsin-tdTomato using a modified calcium phosphate protocol. Briefly, 1 μg of plasmid/well was diluted in 15 μl of water, mixed with 15 μl of 2X HBS (containing 280 mM NaCl, 1.5 mM Na$_2$HPO$_4$, 50 mM HEPES, pH = 7.10) and incubated for 10 min. This solution was added to the neuronal cultures (30 μl/well) and incubated for 30 min. Cells were fixed with 4% PFA at DIV12-14, mounted onto a superfrost slides and imaged with Nikon confocal using a 10x dry objective. The length of dendrites and axons was analysed with Metamorph software. For sparse transfection of Cre or ΔCre recombinase (*Figure 4E–F*), neurons were transfected with the same protocol at DIV4, using 0.5 μg of AAV-syn-EGFP-CRE or AAV-syn-EGFP-ΔCRE plasmids.

## Preparation of neuronal cell lysates and immunoblot

Neurons were lysed directly in sample buffer and used for immunoblotting experiments. Proteins were separated by SDS-PAGE using 4–20% mini protean TGX precast gels (Bio-Rad). Proteins were transferred onto nitrocellulose membranes for 10 min at 2.5 V using the Trans-blot turbo transfer system (Bio-Rad). Membranes were blocked in Tris-buffered saline (5% no-fat milk powder, 0.1% Tween20) for 1 hr at room temperature. Primary antibodies were diluted in the same buffer and incubated overnight at 4°C. The following antibodies were used at 1:1000 dilution: PTPRS (PAC9986, RRID: AB_2802087), HA (901515, Biolegend, RRID: AB_2565334), PAN-RPTP (75–194, Neuromab, RRID: AB_2174700), Nrxns (G394, RRID: AB_2800397), CASK (75–000, Neuromab, RRID: AB_2068730), RBP2 (4193, RRID: AB_2617050), RIM (U1565, RRID: AB_2617054), Liprin (4396, RRID: AB_2617056), Bassoon (141021, Synaptic Systems, RRID: AB_2066979), Velis (U049, RRID: AB_2802084), Mint (P932, RRID: AB_2802085), Synaptobrevin 2 (104211, Synaptic Systems, RRID: AB_887811), Synaptophysin (101011, Synaptic Systems, RRID: AB_887824), Syntaxin 1A (HPC-1, RRID: AB_887843), Synaptotagmin 1 (105011, Synaptic Systems, RRID: AB_887832), Complexin 1–2 (L669, RRID: AB_2802086), GluN1 (114 011, Synaptic systems, RRID: AB_887750), GluN2A (A6473, Invitrogen, RRID: AB_10376044), GluN2B (75–097, Neuromab, RRID: AB_10673405), GluA1 (182003, Synaptic Systems, RRID: AB_2113441), GluA2 (182103, Synaptic Systems, RRID: AB_2113732), PSD-95 (75–028, Neuromab, RRID: AB_2292909), PAN-SHANK (75–089, Neuromab, RRID: AB_10672418), Tubulin (T2200, Sigma, RRID: AB_262133). Combinations of the following IRDye secondary antibodies were used (1:10.000 dilution): IRDye 800CW donkey anti mouse (926–32212, RRID: AB_621847), IRDye 680LT donkey anti mouse (926–68022, RRID: AB_10715072), IRDye 800CW donkey anti rabbit (926–32213, RRID: AB_621848), IRDye 680LT donkey anti rabbit (926–68023, RRID: AB_10706167), from LI-COR. Pseudo colours were then applied to the signals. Detection of the signal was obtained by Odyssey CLx imaging systems (LI-COR). Quantification was performed with Image Studio 5.2 free software.

## Real time-PCR

Total mRNA was extracted from mouse tissues using the RNeasy Plus Mini Kit (Qiagen, Germany) according to manufacturer instructions. PTPRS, PTPRF and PTPRD transcript levels were measured by RT-PCR using the following predesigned FAM-dye coupled detection assays obtained from Integrated DNA Technologies (IDT, Coralville, IO, USA): Mm.PT.58.5137799, Mm.PT.58.45964964, Mm.PT.58.14060589. Mouse GAPDH (4352932E, Applied Biosystems, Warrington, UK) was used as internal control. The qPCR assay was performed using the TaqMan Fast Virus 1-Step Master Mix (Thermo Scientific).

## Immunochemistry

For immunostaining in neuronal cultures, cells were fixed with 4% PFA, washed in PBS, permeabilized with 0.1% TritonX-100 in PBS for 3 min, blocked with 5% NGS in PBS for 1 hr and incubated with primary antibodies overnight at 4°C (anti-vGluT1, AB5905 Millipore, RRID: AB_2301751; VGAT, AB5062P, Millipore, RRID: AB_2301998; MAP2, CPCA-MAP2, Encor Biotechnology, RRID: AB_2138173). Cells were then washed and incubated with secondary antibodies (1:500, Alexa 488, 545, 633, Invitrogen) and mounted with Southern biotech mounting media. Serial confocal z-stack images (1 μm intervals at 2048 × 2048 resolution) were acquired using a Nikon confocal microscope (A1Rsi) with a 60x oil objective (PlanApo, NA1.4). All acquisition parameters were kept constant among different conditions within experiments.

## Stereotaxic injections

AAV viruses (AAD(DJ) serotype) were produced as previously described and injected in P1 pups using a glass micropipette attached to a 10 μl Hamilton syringe (Model 1801N). To target the dorsal CA3, the following coordinates were used: 0.9 mm anterior from the lambda, 1.3 mm from the midline and 0.9–1.5 mm below the scull. Whole cells voltage clamp recordings in acute hippocampal slices were performed at P30 to P35.

## Preparation of acute brain slices for electrophysiology

Acute coronal brain slices containing the dorsal hippocampus were prepared from P30-35 LAR-RPTP triple cKO (LAR-RPTP cTKO) or BL6 mice. Isofluorane-anethesized mice were decapitated, their brain removed and trimmed, and placed in an ice-cold oxygenated (95% $O_2$, 5% $CO_2$) cutting solution that contained (in mM): 228 sucrose, 26 $NaHCO_3$, 11 glucose, 2.5 KCl, 1 $NaH_2PO_4$, 7 $MgCl_2$, 0.5 $CaCl_2$. 250μm-thick slices were cut with a Leica vibratome (VT1200S) and recovered for 30 min at 32°C in oxygenated ACSF solution containing (in mM): 119 NaCl, 2.5 KCl, 1.3 $MgCl_2$, 2.5 $CaCl_2$, 11 glucose, 1 $NaH_2PO_4$, and 26 $NaHCO_3$. Brain slices were then moved to a holding chamber filled with oxygenated ACSF at room temperature for 30 min-1 h.

## Electrophysiological recordings in neuronal cultures and brain slices

For recordings from cultured neurons, dissociated hippocampal neurons were cultured on coverslips and placed in the recording chamber between DIV12-16. For recordings from dorsal hippocampal slices, an incision was made between the CA3 and the CA1 region and the slices were moved to the recording chamber mounted onto an Axioskop FS-2 upright microscope (Zeiss). The microscope was equipped with DIC and fluorescence filters, and a LED source connected to the back port of the microscope via an optic fiber. Both cell culture and brain slices were maintained at ~32°C via a dual-T344 temperature controller (Warner Instruments). Brain slices were continuously perfused with normal oxygenated ACSF (at about 1 ml/min perfusion rate). Electrical signals were recorded at 25 kHz with a two channel Axoclamp 700B amplifier (Axon Instruments), digitalized with a Digidata 1440 digitizer (Molecular devices) that was in turn controlled by Clampex 10.7 (Molecular Devices).

Synaptic currents were recorded using a pipette solution that contained (in mM): 135 Cs methanesulfonate, 8 NaCl, 10 HEPES, 2 ATP-Mg and 0.3 GTP-Na, 0.1 spermine, seven phosphocreatine, 0.3 EGTA, and 5 QX314 (300 mOsm l−1, pH 7.3 adjusted with CsOH), and an external solution (standard ACSF) that contained (in mM): 119 NaCl, 2.5 KCl, 1.3 $MgCl_2$, 2.5 $CaCl_2$, 11 glucose, 1 $NaH_2PO_4$, and 26 $NaHCO_3$. The following pharmacological agents were used: picrotoxin (100 μM, $GABA_A$R blocker, Tocris Bioscience; used in *Figure 3A–C*, *Figure 4A–F Figure 4I–K*, *Figure 6B–E*, *Figure 7A–C*, *Figure 6—figure supplement 1B–C*); CNQX (10 μM, AMPAR blocker, Tocris Bioscience; used in *Figure 3D–F*, *Figure 4C–F*, *Figure 4I–K*), AP5 (50 μM, NMDAR blocker, Tocris Bioscience; used in *Figure 3D–F*, *Figure 3I–K*), TTX (1 μM, voltage gated sodium channel blocker, American Radiolabeled chemicals, for *Figure 3A–F*, *Figure 4G–H*), Ifenprodil (3 μM, GluN2B blocker, Tocris Bioscience; used in *Figure 4I–K*).

AMPA-receptor-mediated EPSCs were recorded at holding potentials of −70 mV, whereas NMDA-receptor-mediated EPSCs were recorded at +40 mV and quantified at the peak in recordings from cultured neurons (*Figure 4D,F*), or at 50 ms after the stimulus artefact in recordings from acute slices (*Figures 6D* and *7B*).

For the experiments in *Figure 4G*, ACSF containing 1 μM AMPA, 100 μM CTZ (Tocris Bioscience), and 1 μM TTX was perfused into the recording chamber containing ACSF with 1 μM TTX. For the experiments in *Figure 4H*, ACSF containing 10 μM NMDA, 10 μM glycine, and 1 μM TTX was perfused into the recording chamber containing ACSF with 1 μM TTX. NMDA-receptor-mediated currents were recorded at +40 mV and measured at the peak.

For optogenetic experiments, EPSCs in *Figure 7A–C* were evoked by 1 ms blue light pulses and recorded in presence of 1 μM TTX and 1 mM 4-AP in the extracellular solution.

Experiments in *Figure 6B* were performed in ACSF with 0 mM $Ca^{2+}$ and 3 mM $Sr^{2+}$ to induce AMPA-receptor-mediated 'asynchronous' EPSCs upon stimulation of Schaffer-collateral inputs 100 μM PTX was added to the bath to block $GABA_A$-receptors). The peak amplitudes of unitary EPSCs were measured between 50–500 ms after the stimulus artefact.

For mIPSC recordings in Figure 3D, the following internal solution was used (in mM): 135 $CsCl_2$, 10 HEPES, 1mM EGTA, 4 ATP-Na and 0.4 GTP-Na (300 mOsm l−1, pH 7.3 adjusted with CsOH).

## Data analysis and statistics

Electrophysiological data were analysed using Clampfit 10.4 (Molecular Devices) or Igor Pro 4.07 (WaveMetrics, Lake Oswego, OR). Statistical analysis was done using the GraphPad Prism software.

## Acknowledgements

We thank Dr. Taulant Bacaj for help with the design of the PTPRF cKO strategy and Dr. Caiying Guo (Janelia) for generating the PTPRF cKO mouse line. We thank the Welcome Trust Sanger Institute Mouse Genetics Projects and its funders for providing the mutant mouse lines for PTPRS and PTPRD. We also thank the Italian Institute of Technology. This research was supported by a grant from the NIMH (MH052804 to TCS), and by the Marie Skłodowska-Curie grant/Minded project (agreement No 754490 in support to AS).

## Additional information

### Funding

| Funder | Grant reference number | Author |
| --- | --- | --- |
| National Institute of Mental Health | MH052804 | Thomas C Südhof |
| H2020 Marie Skłodowska-Curie Actions | 754490 | Alessandra Sclip |

The funders had no role in study design, data collection and interpretation, or the decision to submit the work for publication.

### Author contributions

Alessandra Sclip, Data curation, Formal analysis, Validation, Investigation, Methodology, Designed the experiments, Performed the experiments, and Analyzed the data and wrote the manuscript; Thomas C Südhof, Funding acquisition, Designed the experiments, Performed the experiments, and Analyzed the data and wrote the manuscript

### Author ORCIDs

Alessandra Sclip (iD) https://orcid.org/0000-0002-9313-4176

### Ethics

Animal experimentation: Animal experiments were conducted following protocols approved by the Administrative Panel on Laboratory Animal Care at Stanford University Protocol number APLAC-20787.

### Decision letter and Author response

Decision letter https://doi.org/10.7554/eLife.53406.sa1
Author response https://doi.org/10.7554/eLife.53406.sa2

## Additional files

### Supplementary files
• Transparent reporting form

### Data availability

All data generated or analysed during this study are included in the manuscript and supporting files.

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
