## [Decision Letter]

**Acceptance summary:**

The authors test the function of LAR-RPTPs by conditional deletion of all RPTP isoforms, first in neuronal cell culture and then in vivo in the hippocampal network where these isoforms are known to be expressed. Despite a large developmental literature on the function of these intriguing molecules, the authors discover no major anatomical deficits when all isoforms are removed. Rather, both in vitro and in vivo, the authors describe diminished NMDA receptor abundance, assessed primarily through electrophysiological synaptic recordings. This finding is important and of broad relevance to the field of developmental neuroscience.

**Decision letter after peer review:**

Thank you for submitting your article "LAR receptor phospho-tyrosine phosphatases regulate NMDA-receptor responses" for consideration by *eLife*. Your article has been reviewed by three peer reviewers, including Graeme W Davis as the Reviewing Editor and Reviewer #1, and the evaluation has been overseen by Gary Westbrook as the Senior Editor. The reviewers have discussed the reviews with one another and the Reviewing Editor has drafted this decision to help you prepare a revised submission.

Summary:

This is a straightforward and definitive study from Sclip and Sudhof. The findings are important and of broad relevance to the field of developmental neuroscience, and may well lead to interesting new experiments in the realm of synaptic plasticity, learning and behavior in the future. I advise against requiring such experiments in the current manuscript, which already presents a thorough characterization and primary observation that can be built upon in the future.

Essential revisions:

The only major comment concerns suggested text revisions.

Although the data establish that synaptic NMDAR responses are decreased, it remains uncertain that this effect is mediated by re-localization of synaptic NMDARs to extrasynaptic sites. One alternative explanation is that RPTPs trans-synaptically modulate NMDAR properties, rather than their localization. The best solution to solve this point would be to remove the conclusion from the impact statement and Abstract, and to provide instead a balanced presentation of how NMDAR responses may be modulated by RPTPs in the Discussion section. In my view, this would not lower the overall impact of the study because the main conclusion is that IIa RPTPs on their own do not control synaptogenesis.

---

## [Author Response]

Essential revisions:The only major comment concerns suggested text revisions.Although the data establish that synaptic NMDAR responses are decreased, it remains uncertain that this effect is mediated by re-localization of synaptic NMDARs to extrasynaptic sites. One alternative explanation is that RPTPs trans-synaptically modulate NMDAR properties, rather than their localization. The best solution to solve this point would be to remove the conclusion from the impact statement and Abstract, and to provide instead a balanced presentation of how NMDAR responses may be modulated by RPTPs in the Discussion section. In my view, this would not lower the overall impact of the study because the main conclusion is that IIa RPTPs on their own do not control synaptogenesis.

We changed the Abstract and the impact statement according to the suggestion of the reviewers.

Abstract:

“LAR-type receptor phosphotyrosine-phosphatases (LAR-RPTPs) are presynaptic adhesion molecules that interact trans-synaptically with multitudinous postsynaptic adhesion molecules, including SliTrks, SALMs, and TrkC. […] Thus, LAR-RPTPs are not essential for synapse formation, but control synapse properties by regulating postsynaptic NMDA-receptors via a trans-synaptic mechanism that likely involves binding to one or multiple postsynaptic ligands.”

Impact statement:

“LAR-RPTPs are not essential for synapse formation, but they are important determinants of synapse properties as they contribute to regulating postsynaptic NMDA receptor function.”